# Utilizing Wheel Washing Machine Sludge as a Cement Substitute in Repair Mortar: An Experimental Investigation into Material Characteristics

**DOI:** 10.3390/ma17092037

**Published:** 2024-04-26

**Authors:** Changhwan Jang, Tadesse Natoli Abebe

**Affiliations:** 1Department of Smart Construction and Environmental Engineering, Daejin University, 1007 Hoguk-ro, Pocheon-si 11159, Republic of Korea; cjang@daejin.ac.kr; 2Civil and Environmental Engineering Department, Hanyang University, Jaesung Civil Engineering Building, 222 Wangsimni-ro, Seongdong-gu, Seoul 04763, Republic of Korea

**Keywords:** wheel cleaning sludge, repair mortar, pretreatment, additive material, substitution

## Abstract

The construction industry strives for sustainable solutions to tackle environmental challenges and optimize resource use. One such focus area is the management of industrial byproducts and waste materials, including fugitive dust control through wheel washers. While wheel washers play a pivotal role in dust management, they generate a challenging byproduct known as wheel washer sludge (WWS). The disposal of WWS is complicated due to its composition and potential hazards. Recent research explores reusing WWS in construction materials, particularly in repair mortar, aiming for sustainability and circular economy principles. This study investigates the incorporation of WWS into repair mortar formulations, evaluating mechanical properties, durability, and environmental implications. Results show that WWS enhances workability but prolongs setting time. Mechanical strength tests reveal improvements with WWS addition, especially when pretreated. Water absorption rates decrease with pretreated WWS, indicating enhanced durability. Chemical attack tests demonstrate resistance to carbonation and chloride penetration, especially with modified WWS. Freeze–thaw tests reveal improved resistance with WWS addition, particularly modified. Microstructure analysis confirms hydration products and denser matrices with WWS inclusion. Environmental hazard analysis shows WWS contains no harmful heavy metals, indicating its suitability for use in repairs. Overall, this study highlights the technical feasibility and environmental benefits of incorporating WWS into repair mortar, contributing to sustainable construction practices.

## 1. Introduction

The construction industry is constantly seeking innovative solutions to address environmental sustainability challenges and enhance resource efficiency. One area of focus is the effective management and utilization of industrial byproducts and waste materials, aiming to reduce environmental impacts while promoting circular economy principles [1,2]. Among the myriad challenges faced by construction sites, one often overlooked yet critically important concern is the management of fugitive dust—a problem addressed through the ubiquitous presence of wheel washers [3,4,5]. These machines stand as silent sentinels at the gateways of construction sites, tasked with the vital role of cleansing the tires of vehicles as they enter and exit, thereby curbing the spread of dust and maintaining the cleanliness of surrounding roads. Embedded within the fabric of construction sites, wheel washers serve as a frontline defense against fugitive dust. This duty is not merely recommended but mandated by regulations such as the Enforcement Rules of the Air Quality Conservation Act. This legal framework underscores the indispensable nature of wheel washers in preserving air quality and upholding environmental standards [6,7,8,9,10,11]. However, beyond their role in dust control, these machines also generate a byproduct sludge that poses a unique challenge in terms of disposal and management [12].

As construction vehicles traverse through wheel washers, the bleeding of contaminated soil and water gives rise to sludge, categorized as construction waste under prevailing regulations. Yet, the disposal of this sludge presents a multifaceted dilemma. Laden with moisture and potentially hazardous components such as vehicle oil and various waste materials, wheel washer sludge (WWS) defies easy disposal methods [13,14]. Regrettably, due to a lack of awareness among site officials, much of this sludge finds its way into improper disposal channels, posing risks of soil contamination and environmental pollution. However, recent research has explored the possibility of reusing WWS as a viable alternative to conventional construction materials, particularly in cement-based applications such as repair mortar [15,16,17]. Cement production is known for its substantial environmental footprint, contributing to carbon emissions, energy consumption, and depletion of natural resources. As such, there is growing interest in developing sustainable alternatives to traditional cementitious materials [18,19,20]. By substituting cement with waste-derived materials like WWS, not only can the environmental burden associated with cement production be mitigated, but also the utilization of industrial byproducts can be optimized, fostering a more circular approach to construction materials management. WWS exhibits a mineralogical composition akin to that of clay and Portland cement, characterized by the presence of significant oxides, including SiO_2_, Al_2_O_3_, CaO, and Fe_2_O_3_ in a range between 40–50%, 20–25%, 2–5%, and 4–7%, respectively. Leveraging its chemical constituents, sewage sludge finds extensive application in the manufacturing of construction and building materials, such as eco-cement, bricks, ceramic materials, and lightweight aggregates (LWAs), as well as supplementary cementitious materials (SCMs) [9,21].

Repair mortar is crucial in maintaining and rehabilitating concrete structures, offering solutions for structural integrity, durability, and aesthetic appeal. Traditionally composed of cement, aggregates, and water, the performance of repair mortar can be influenced by various factors, including material composition, mixing proportions, and curing conditions [22,23]. Introducing alternative materials such as WWS into repair mortar formulations presents an opportunity to enhance its properties while reducing reliance on virgin resources. The feasibility of incorporating WWS into repair mortar hinges upon a comprehensive understanding of its material characteristics and its interaction with other components in the mortar matrix. Previous studies have investigated the physicochemical properties of WWS, highlighting its potential as a supplementary cementitious material (SCM) due to its pozzolanic or cementitious properties [24,25]. However, limited research has been conducted to evaluate the performance of repair mortar formulations containing WWS, particularly concerning its mechanical properties, durability, and long-term behavior.

The primary objective of this research is to delve deeply into the potential integration of WWS as a constituent in repair mortar, aiming for future applicability. Through a partial replacement of cement with WWS, a novel environmentally sustainable cement mortar was formulated. Testing specimens were meticulously prepared by substituting WWS cement at varying percentages (one percent, ten percent, and twenty percent by weight) in both the pre and post-ball milling process. This study meticulously examined the mechanical attributes of the resulting cement mortar containing WWS and compared them against a control group devoid of WWS content. 

Through adherence to standard specifications and rigorous testing protocols, including assessments of compressive and flexural strength, water absorption, and durability aspects, this study unveiled the potential of WWS as a viable cement replacement in cement mortar fabrication. This comprehensive approach encompassed the characterization of input materials and a thorough investigation into the physio-mechanical properties of the resultant products. Moreover, the research advocates for an environmental hazard analysis to further elucidate the sustainability implications of WWS integration. In summary, this study not only explores the technical feasibility of incorporating WWS into cement mortar but also underscores its environmental benefits and potential hazards, contributing to a holistic understanding of its applicability in construction materials. 

## 2. Experimental Procedure

### 2.1. Raw Materials

The experimental investigation employed ordinary Portland cement (OPC) as the principal binding agent. Table 1 presents the chemical and physical characteristics of OPC as per the specifications outlined in ASTM C150 [26]. Fine aggregates, specifically natural sand, with a maximum size of 4.75 mm conformed to the grading criteria for concrete aggregates outlined in ASTM C33-2003 [27]. Wheel washer sludge, both untreated and treated, was employed as a supplementary cementitious material. Comprehensive analyses encompassing chemical, mineralogical, and physical properties such as specific gravity, surface area, and particle size distribution were conducted, with the findings summarized in Table 2 and Figure 1 and Figure 2, respectively. To enhance the workability of the cement mortar, a polycarboxylic water reducer was integrated, with its physical and chemical attributes detailed in Table 3.

### 2.2. Mix Proportion and Specimen Preparation

This study involved the preparation and analysis of five sets of mortar specimens with varying mixing proportions, as illustrated in Table 4. Waste wood ash (WWA) was incorporated as a supplementary cementitious material (SCM) at replacement levels of 10% and 20%. The water-to-cement ratio was maintained at 0.4, and a yellow-colored polycarboxylic water-reducing admixture, comprising 0.7% of the mixture, was added to enhance the workability of the fresh mortar [28]. Before the addition of water, a dry mixing of cement and aggregate was carried out for 3 min. Subsequently, water and the water reducer were added, followed by an additional 5 min of mixing. The fresh mixtures were then compacted into molds through tamping. After a 24-h air curing period, the specimens were demolded and immersed in water at a constant temperature of 23.2 °C for the subsequent 28 days. 

The following specimens were made for each mix design proportion, as presented in Table 4. The specimen geometry varied depending on the specific test conducted following the standard. Flow tests and setting time tests were exceptions, as they were performed on fresh cement mortar. For microstructural characterization through X-ray diffraction (XRD) analysis and environmental hazard analysis, powder samples with a particle size smaller than 75 µm were obtained by grinding specimens using standard sieve sizing. These samples were derived from the crushed specimens utilized in mechanical strength testing. Furthermore, for scanning electron microscopy (SEM) analysis, samples of approximately 0.5 cm in dimension were carefully extracted from the inner core of the specimen. Before testing, these samples were meticulously stored in isopropanol to prevent potential hydration processes.

### 2.3. Experimental Methods and Methodology 

#### 2.3.1. Flow Tests and Setting Time

The flow test, conducted according to ASTM C 1437 standards [29], utilized a standard flow mold with open upper and lower diameters of 70 mm and 100 mm, respectively, and a height of 50 mm. The mold was centered on the flow table, and a 25 mm-thick first mortar layer was compacted 20 times. A second layer was then added until overflow, leveled, and the mold removed. The table was dropped 25 times within 15 s, and the diameter of the fresh mortar was measured along four guidelines, with the flow spread diameter calculated as their average. To determine the workability of the repair cement mortar, the setting time was measured according to the method provided in the literature. Figure 3 shows the apparatus used for performing the aforementioned tests.

#### 2.3.2. Mechanical Strength Test 

An extensive experimental study was conducted to assess the mechanical strength of cubic and prismatic specimens. Cubic specimens measuring 50 × 50 × 50 mm^3^ underwent compressive testing before and after electric stress application using a Shimadzu CCM-200A universal testing machine (Shimadzu, CCM-200A, Shimadzu Corporation, Kyoto, Japan) with a 200-ton capacity, following ASTM C109 standards [30]. Flexural strength was evaluated on 40 × 40 × 160 mm^3^ prismatic specimens using a Shimadzu AG-I 250 KN flexural bending machine (Shimadzu, AG-I 250 KN, Shimadzu Corporation, Kyoto, Japan) with a 25-ton capacity, following ASTM C348 guidelines [31]. Five specimens were prepared for each mix proportion to ensure result reliability, and the mean strength value was calculated for each set.

#### 2.3.3. Water Absorption Rate Test

The flow test, conducted according to ASTM C642-21 standards [32], assesses the flow properties of a material. Water absorption refers to a material’s capacity to absorb and retain water, assessed by measuring the water saturation of a specimen. Test samples were dried at 105 °C until reaching a constant weight (*m*_1_), then submerged in water at (20 ± 2) °C with the water level positioned 50 mm above the sample. After 24-h intervals, samples were weighed under air conditions with 0.1% accuracy (*m*_2_). Saturation was determined when the difference between consecutive weighings did not exceed 0.1%. Water absorption (*W_abs_*) was calculated accordingly:(1)$$W_{abs}=\frac{m_{2}-m_{1}}{m_{1}}\cdot 100\%,$$
where *W_abs_* is the water absorption in percentage, *m*_2_ is the mass of the test sample after saturation, and *m*_1_ is the mass of the air-dried test sample.

#### 2.3.4. Chloride Penetration Test

The chloride penetration test, conducted following NT-BUILD 492 [33], as shown in Figure 4, utilized cylindrical specimens with a height of 50 mm and a diameter of 100 mm. Before the experiment, specimens were immersed in a Ca(OH)_2_ solution for 18 h. Testing lasted 24 h under constant temperature and humidity conditions, applying 30 V and 0.46 mA for WWS specimens. Following the experiment, specimens were air-dried for 1 day and split hydraulically. Chlorine penetration length was determined by applying AgNO_3_ solution to the split surface, disregarding a 10 mm section on both sides to mitigate error. Measurements were averaged from seven lengths, spaced at 10 mm intervals on a smoother surface.

#### 2.3.5. Carbonation Depth Test

Specimens were cast following ISO 1920-12:2015 [34]. After 24 h of casting, 120 specimens were demolded for each concrete grade and cured at (20 ± 1) °C and (95 ± 5)% relative humidity for 28 d. Subsequently, specimens were dried at 60 °C for 48 h and placed in an environmental simulation test chamber for a week to equilibrate temperature and humidity with the external environment as shown in Figure 5a,b. To accelerate carbonation testing, a high CO_2_ concentration was applied, accelerating the carbonation method. The environmental simulation test system provided conditions for carbonation testing. Carbonation depth was assessed using 1% phenolphthalein reagent in 95% ethyl alcohol. Ten points on the specimen’s cross-section were measured for carbonation depth, with an accuracy of 0.1 mm, and their average value was determined.

#### 2.3.6. Freeze–Thaw Test

In this experimental study, to evaluate the durability of repair materials containing wheel wash sludge, the thermal cycling protocol necessitates adherence to specific temperature parameters whereby the core temperature of the specimens is maintained within the range of −18 ± 2 °C to 6 ± 2 °C for 4 h per cycle. This cycle has iterated a total of 300 times, with measurements of the Relative Dynamic Modulus (RDM) and mass loss conducted every 30 cycles. The experimental procedure conforms to the guidelines outlined in the ASTM standard [35], with RDM calculated according to Equation (2):(2)$$P_{r}=\frac{n_{m}^{2}}{n_{0}^{2}}\times 100$$
where *Pr* represents the relative dynamic modulus (%), *n_m_* denotes the measured frequency after *m* cycles of Fourier transform (FT), and *n*_0_ signifies the measured frequency at the initiation of FT cycling.

#### 2.3.7. Microstructure Analysis

The crystallographic structure of both the cement mortar and conductive mortar was analyzed through X-ray diffraction (XRD) using a Goniometer Ultima+ instrument from Mitsubishi Tanabe Pharma, Osaka, Japan. XRD measurements were performed with a wavelength of 1.54 Å, scanning within a 2θ range of 10–65°, at a speed of 2°/min, and with an applied voltage of 40 kV at 30 mA. The compositional and morphological characteristics of the WWS and its distribution throughout the matrix were observed using SEM, as performed in previous studies [36].

#### 2.3.8. Environmental Hazard Analysis

In this experimental investigation, the impact of wheel wash sludge, a field-generated waste, was deemed necessary to analyze. An environmental assessment was conducted to identify hazardous substances, utilizing ICP-MS analysis for detection. ICP-MS (Inductively Coupled Plasma Mass Spectrometer) operates by detecting ions through inductively coupled plasma, as depicted in Figure 6. With a detection limit in the parts per billion (ppb) range, ICP-MS enables accurate qualitative and quantitative analysis of trace elements simultaneously.

## 3. Results and Discussion

### 3.1. Flow TESTS and Setting Time

The impact of both treated and untreated wheel wash sludge (WWS) on the workability of WCS and M-WCS was evaluated through the slump test, as depicted in Figure 7. The results indicate a consistent enhancement in a slump with an increase in WWS content. Compared to the CS, WCS exhibits a slump increase of 2.4% and 4.3% for cement substitution levels of 10% and 20%, respectively. Conversely, for M-WCS, the slump slightly decreases by 1.2% and 2.6% for the same cement substitution levels. This variation in trend is attributed to the higher particle size distribution in M-WCS, as shown in the previous studies. Ball milling is attributed to a smoother surface and higher water absorption, which increases the friction between the aggregate particles and consequently restricts the flow of fresh concrete [37,38]. In addition, it impedes the steric repulsion effect of the superplasticizer. In contrast, WCS enhances workability due to its greater paste volume, which results from its lower specific gravity compared to CS. The increased paste volume contributes to additional lubrication effects in fresh concrete, reducing the energy required for compaction or flow enhancement [39,40]. 

The results depicted in Figure 8 demonstrate that the substitution of cement with WCS significantly delays both initial and final setting times, with the setting time prolonging at higher levels of cement substitution. Notably, Figure 8 illustrates both linear and non-linear relationships between setting time and WCS cement substitution. This extension in setting time may be attributed to the reduced rate of ettringite production resulting from gypsum reaction with water, as well as the hydration reaction generating a skeletal framework (C-S-H) from the reactions of alite (C_3_S) and belite (C_2_S). These findings align with previous research [41] on High-Strength Concrete (HSC), which incorporates various supplementary cementitious materials. Furthermore, the delay in setting time may be attributed to the decrease in portlandite generation from hydration reactions with increasing cement content, while the rate of subsequent pozzolanic reactions remains constant. Conversely, Modified Waste Ceramic Slurry (M-WCS) exhibited a slight reduction in setting time.

### 3.2. Mechanical Strength

The compressive strength characteristics of WWS before and after pretreatment were analyzed through a series of compressive strength tests in this experimental study. Figure 9a presents the results of these tests for samples containing WWS. For the general repair material, compressive strength reached 28.06 MPa after 3 days, increasing to 35.57 MPa and 42.17 MPa after 7 and 28 days, respectively. Samples containing 10% WCS without pretreatment exhibited strengths of 24.51 MPa after 3 days, 37.45 MPa after 7 days, and 44.11 MPa after 28 days. The 20% WCS sample showed 20.94 MPa after 3 days, 35.06 MPa after 7 days, and 41.66 MPa after 28 days, indicating a slight improvement in strength compared to the general sample. Samples treated with M-WCS displayed enhanced strength, with the 10% M-WWS sample achieving 24.74 MPa after 3 days, 41.66 MPa after 7 days, and 49.28 MPa after 28 days, representing over a 15% improvement compared to the general sample. The 20% M-WCS sample showed 21.46 MPa after 3 days, 36.27 MPa after 7 days, and 43.19 MPa after 28 days, indicating a 3% improvement compared to the control sample. At 7 days, the addition of WCS was more effective in increasing the compressive strength of the specimen. According to [1,42], construction sludge concrete also exhibited greater strength than control specimens at the same levels of cement substitution at the early ages. The high efficiency of WCS in its early stages may be attributed to its chemical composition, which had a higher concentration of CaO and a lower concentration of SiO_2_ compared to the reference specimen. The presence of CaO in WWS facilitates a quick chemical reaction with water, resulting in the development of strength during the initial stages of the curing process. Furthermore, CaO is essential in promoting the formation of compressive strength in concrete during the initial stages of the curing process [43]. After 28 days, there was an 11% improvement compared to the untreated 10% sample. The increase in strength is attributed to pozzolanic reactions involving portlandite (Ca(OH)_2_) and silica (SiO_2_) compositions of WWS, resulting in the formation of secondary hydration products (C-S-H) at an accelerated rate [44,45].

Furthermore, the bending strength tests, depicted in Figure 9b, demonstrated that the general sample achieved strengths of 5.14 MPa after 3 days, 6.23 MPa after 7 days, and 7.20 MPa after 28 days. The 10% untreated WWS sample displayed strengths of 4.67 MPa after 3 days, 6.28 MPa after 7 days, and 7.23 MPa after 28 days, showing slightly improved strength compared to the general sample. Similarly, the 20% untreated WCS sample exhibited strengths of 4.46 MPa after 3 days, 6.26 MPa after 7 days, and 7.25 MPa after 28 days, displaying a trend similar to the general sample. Conversely, the 10% M-WWS sample showed strengths of 4.93 MPa after 3 days, 6.93 MPa after 7 days, and 8.01 MPa after 28 days, while the 20% M-WWS sample achieved strengths of 4.87 MPa after 3 days and 6.86 MPa after 7 days, with a further increase to 7.96 MPa after 28 days. These results indicate improvements in strength compared to both the general and untreated WCS samples, with an 11% enhancement observed compared to the general sample after 28 days. It is concluded that repair mortar containing WWS is suitable for use.

### 3.3. Water Absorption Test

After 28 days of curing, the water absorption test was conducted on mortar specimens, and the results are presented in Figure 10. The test revealed that the water absorption rate for the control specimen (CS) was 6.5%, while for the specimen with 10% WCS, it was slightly higher at 6.7%, albeit still within acceptable limits indicated by the dotted line on the diagram. However, for specimens containing 20% WCS, the absorption rate was relatively higher at 7.1%. Conversely, Modified Waste Ceramic Slurry (M-WCS) specimens exhibited absorption rates of 6.1% and 6.5% for 10% and 20% substitution levels, respectively, compared to the control specimen. The milling process of M-WCS facilitates a higher surface area for pozzolanic reactions, leading to the formation of additional cementitious compounds over time and resulting in a denser matrix [46,47]. Consequently, the absorption rate was reduced by approximately 6%. In previous studies, a similar trend was observed in the increase in water absorption in WCS-10 and WCS 20, suggesting that beyond the optimal cement replacement, non-treated WWS significantly increases the porosity of repair mortar. In general, the reduction of porosity is accomplished by effectively filling the gaps between bigger particles with smaller particles, as observed in the M-WCS specimen. When there are more smaller particles than needed to fill the gaps, the extra smaller particles might push the larger particles apart, generating bigger empty spaces that let more water in. The moisture content of concrete naturally varies due to the addition of water during the mixing process. However, the process of cement hydration, which involves the drying of mortar and the formation of gaps between particles, leads to increased water absorption, particularly in dried mortar [39,48]. Hence, it was concluded that the use of WWS as a repair material after pretreatment is suitable. 

### 3.4. Chemical Attack Test

Figure 11a,b depict the carbonation depth and chloride penetration depth, respectively. For the control specimen (CS), the carbonation depth was measured at 11.21 mm. Notably, specimens containing 10% and 20% WCS exhibited higher carbonation depths, with increments of 1.7% and 3.2%, respectively. Conversely, M-WCS specimens with 10% and 20% content showed marginal decrements in carbonation depth, with reductions of 9.8% and 2.5%, respectively, indicating higher resistance compared to the general sample. This resistance is attributed to the addition of finer SCMs, which improve packing density and reduce porosity. Moreover, higher incorporation of M-WCS leads to increased formation of calcium silicate hydrate (CSH) products, inhibiting carbonation movements. Therefore, it is recommended that WWS be used after pretreatment for enhanced durability.

In Figure 11b, the results of passing current through concrete samples for 24 h at 28 days of age are presented. The chloride ion permeability levels of the WWS repair mortar samples were considerably lower than that of the CS, indicating good resistance to chloride penetration in WCS blended specimens. This resistance is attributed to the pozzolanic reaction of WCS, which consumes Ca(OH)_2_ produced via cement hydration, resulting in improved Interfacial Transition Zone (ITZ) and blocking of most connected pores by reaction products [49,50]. As provided in the studies [1,51], the resistance of chloride ion penetration is enhanced due to the pozzolanic reaction of WWS in the matrix, creating more CSH, which traps more chloride ions and blocks the ingress path. The results were also in agreement with the findings of previous researchers [52,53], who revealed that the volume of pores in concrete declined with the decrease in particle size. This made concrete more highly resistant to chloride ion penetration.

### 3.5. Freeze–Thaw Damage

Figure 12 presents the variations in the relative dynamic modulus of elasticity of both the control mix and mixtures containing WWS throughout Freeze–Thaw (F–T) cycles. The results indicate a decrease in the relative dynamic modulus of elasticity with an increase in the number of cycles. Notably, the specimens experienced a significant drop in the dynamic modulus of elasticity during the initial F–T cycles, indicating internal damage caused by the formation of microcracks upon exposure to the first F–T cycles. However, the addition of WWS appears to mitigate the impact of freezing and thawing on the dynamic modulus of elasticity, as evidenced by a lesser decrease in modulus with an increasing number of cycles [54,55]. This suggests that the internal damage caused by freezing is reduced through the presence of WWS. This can be attributed to the dense microstructure of cementitious composites incorporating WWS, resulting in reduced permeability and a relative increase in mechanical strength. 

After exposure to 300 cycles, the dynamic modulus of elasticity of the specimen containing 10% M-WCS decreased by only 14%. In contrast, specimens containing 20% WCS and M-WCS exhibited decreases of approximately 24.3% and 22.3%, respectively, compared to the control mix, which experienced a 40% decrease after exposure to 280 cycles, marking the end of the test according to ASTM C666 [35]. Unlike the above test results, in this case, the WCS specimens showed a better performance than M-WCS. This is because freezing and thawing resistance is determined using entrained air and the strength of the cement matrix rather than the packing (filler) effect and contribution of the pozzolanic reaction [37].

### 3.6. Microstructure Analysis

Figure 13 presents the X-ray diffraction (XRD) analysis of representative specimens at a curing age of 28 days, both before and after the application of electric stress. The analysis reveals the presence of various crystalline structures, including quartz, calcium hydroxide (Ca(OH)_2_), calcium carbonate (CaCO_3_), ettringite, dolomite (CaCO_3_·MgCO_3_), and others. When cement mortar is prepared with wheel wash sludge (WWS) as a cement substitution, the XRD pattern displays additional peaks or shifts in peak positions compared to plain cement mortar. These variations can be attributed to the higher SiO_2_ content in WWS, which possesses a crystal structure and lattice parameters different from those of the minerals in cement. At a replacement ratio of 20%, the XRD pattern exhibits a dominant peak corresponding to SiO_2_. However, in WWS-incorporated specimens, hydration products such as calcium hydroxide (CH) at 17.99° and 22.55°, C_3_S at 29.41°, quartz at 26.55°, and calcium silicate hydrate (CSH) at 27.47° were observed in similar quantities.

Figure 14 depicts the Scanning Electron Microscopy (SEM) observation and Energy Dispersive Spectroscopy (EDS) analysis results for a sample containing 20% of untreated and milled WWS, both before and after milling. The results show that after 3 days, a significant number of ettringite and CH products were observed in the untreated sample. Conversely, in the milled sample, hydration products were formed slightly more slowly than in the CS. However, after 28 days, numerous sections displayed C-S-H hydrate. The observation results indicate a very high density of the sample. Additionally, the EDS analysis reveals higher Si and Ca content ratios compared to CS. Despite this, the strength of the specimen after milling treatment was found to be improved. This enhancement can be attributed to the smooth surface texture of M-WCS, which acts as nucleation sites, resulting in a larger volume fraction of these materials being present.

### 3.7. Environmental Hazard Analysis

Given that WWS is a waste product generated in field operations, it is imperative to verify the presence of harmful substances. Specifically, the detection of lead, copper, arsenic, mercury, hexavalent chromium, and cadmium, classified as the six major heavy metals, is crucial due to their significant environmental hazards. However, upon conducting experiments, it was determined that none of these six major heavy metals were detected, as indicated in Table 5. Furthermore, no harmful factors associated with the waste processing test items were detected either. Consequently, it was concluded that wheel washer sludge could be considered a safe material for use in repairs, as no harmful factors were detected.

## 4. Conclusions

The experimental study aimed to produce repair mortar using sludge from a wheel washer as a cement substitute. Pretreatment through ball milling was employed to enhance the sludge’s physical properties. This study assessed the sludge’s usability by comparing its work performance, physical and mechanical properties, durability, and environmental hazards with traditional cement and drew the following conclusions:Incorporating WWS into repair mortar formulations improved workability, as seen in the slump test results. There was a consistent increase in slump, with 10% and 20% WWS substitutions resulting in enhancements of 2.4% and 4.3%, respectively. However, higher WWS content led to a notable delay in both initial and final setting times compared to the control sample.WWS-containing specimens exhibited superior resistance to internal damage from freeze–thaw cycles. For example, after 300 cycles, specimens containing 10% M-WWS showed a decrease in modulus of only 14%, significantly lower than the 40% decrease observed in the control mix. This resilience suggests potential long-term durability in WWS-based repair mortars.

However, the effectiveness of WWS as a cement substitute may vary depending on its origin and composition, necessitating comprehensive characterization and quality control measures. In addition, While the experimental investigation did not detect harmful substances in WWS, ongoing monitoring and analysis are essential to ensure environmental safety and regulatory compliance. Further research is warranted to explore advanced processing techniques for enhancing the performance and sustainability of WWS-based repair mortar formulations.

## Figures and Tables

**Figure 1 materials-17-02037-f001:**
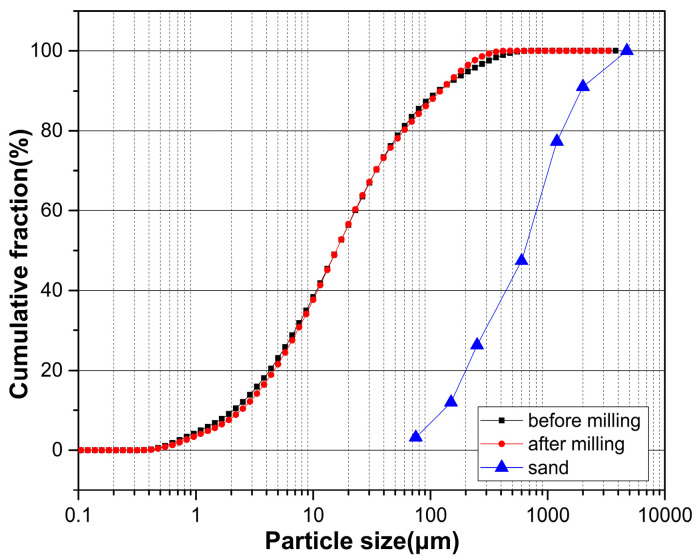
Particle size distribution of the material used.

**Figure 2 materials-17-02037-f002:**
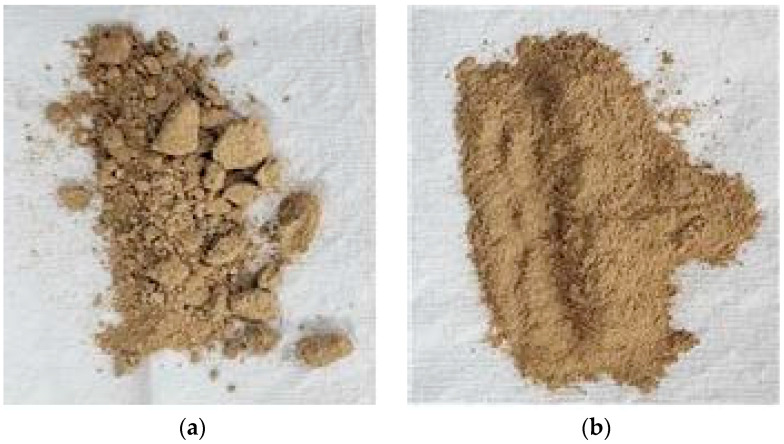
(**a**) Before and (**b**) after milling of WWS.

**Figure 3 materials-17-02037-f003:**
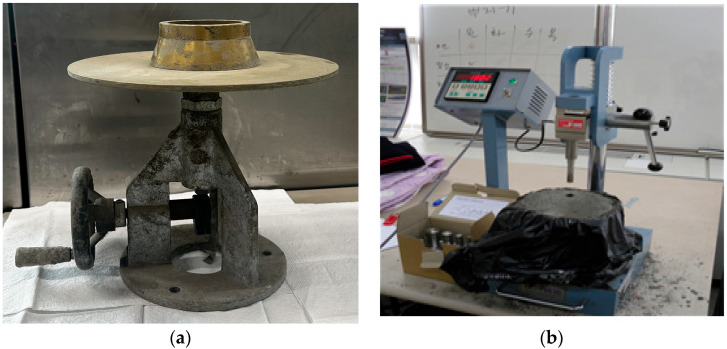
(**a**) Followability test and (**b**) setting time test of WWS mortar.

**Figure 4 materials-17-02037-f004:**
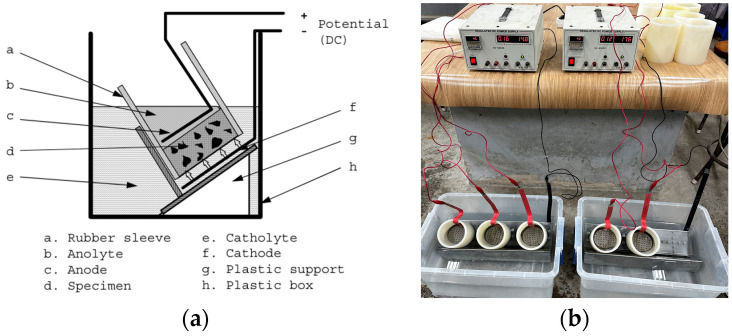
Chloride penetration test: (**a**) NT-Build 492 Standard [33] and (**b**) actual experimental setup.

**Figure 5 materials-17-02037-f005:**
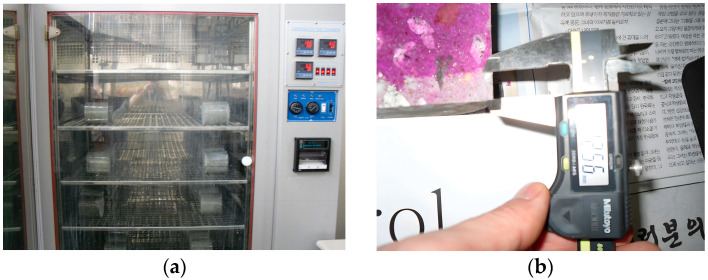
(**a**) Accelerated carbonation chamber and (**b**) carbonation depth measurement.

**Figure 6 materials-17-02037-f006:**
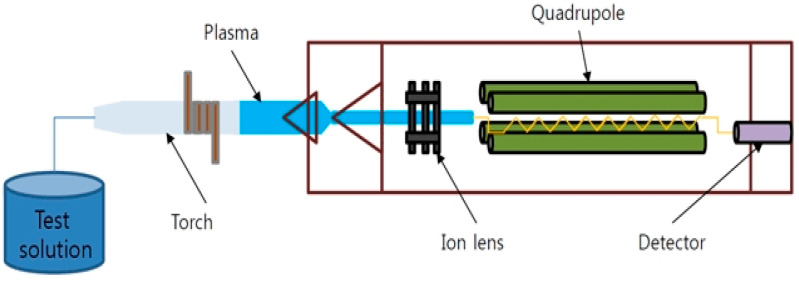
Schematic setup of environmental hazard analysis test.

**Figure 7 materials-17-02037-f007:**
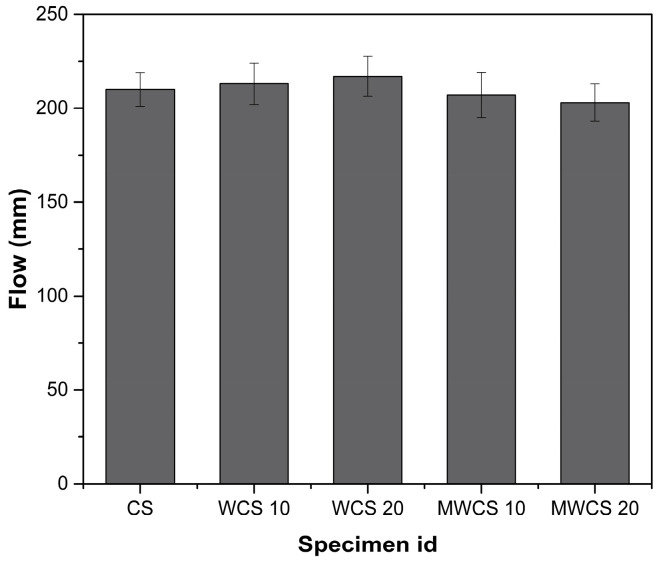
Flow test of mortar.

**Figure 8 materials-17-02037-f008:**
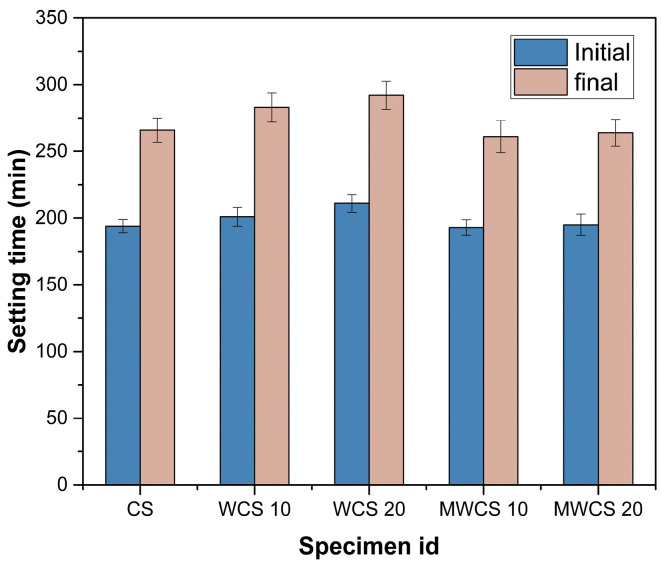
Setting time.

**Figure 9 materials-17-02037-f009:**
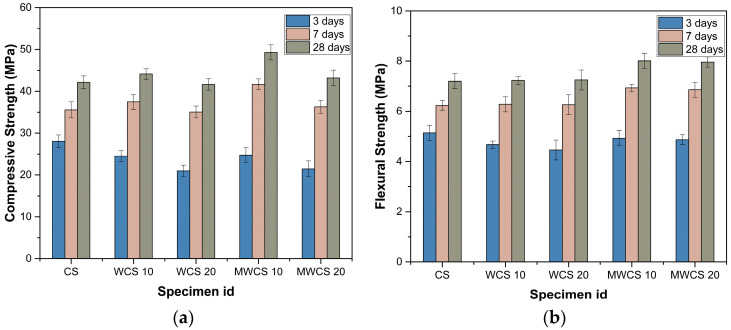
Mechanical properties: (**a**) compressive strength and (**b**) flexural strength.

**Figure 10 materials-17-02037-f010:**
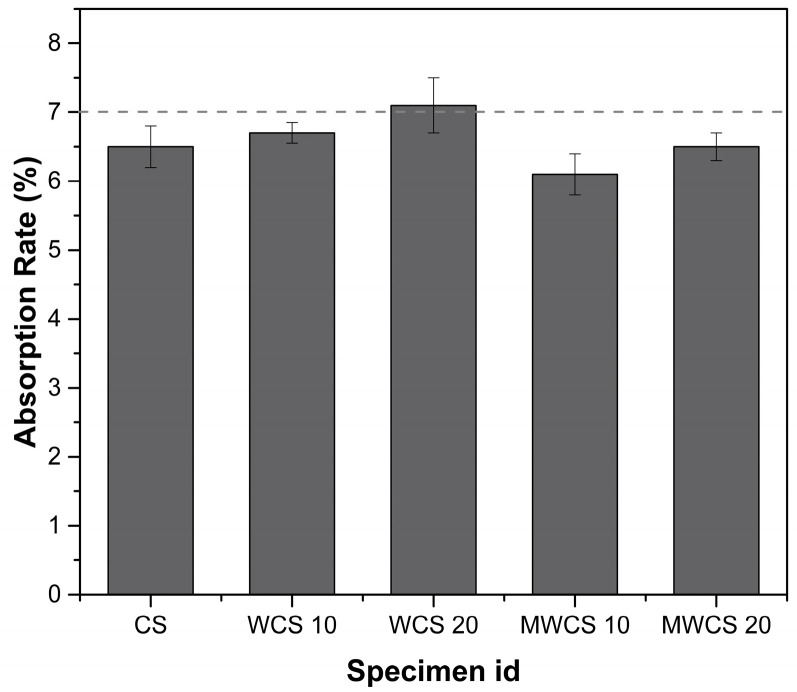
Water absorption rate.

**Figure 11 materials-17-02037-f011:**
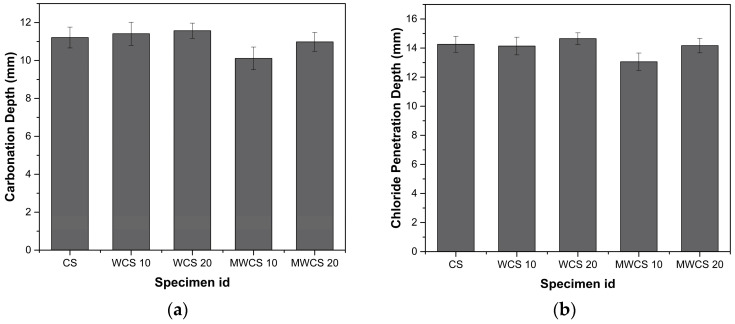
Chemical attack tests: (**a**) carbonation attack and (**b**) chloride penetration depth.

**Figure 12 materials-17-02037-f012:**
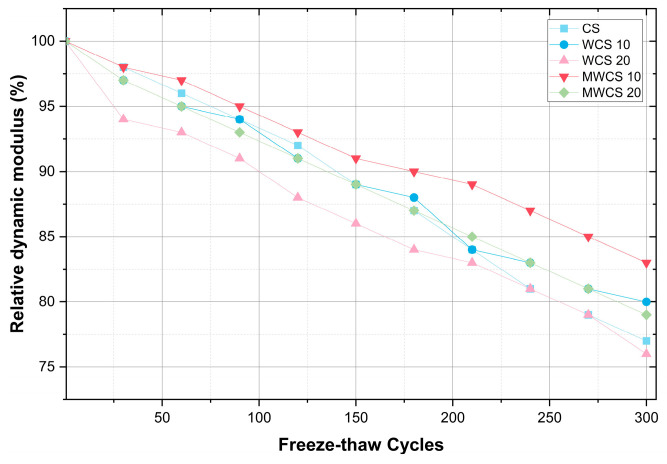
Freeze–thaw damage.

**Figure 13 materials-17-02037-f013:**
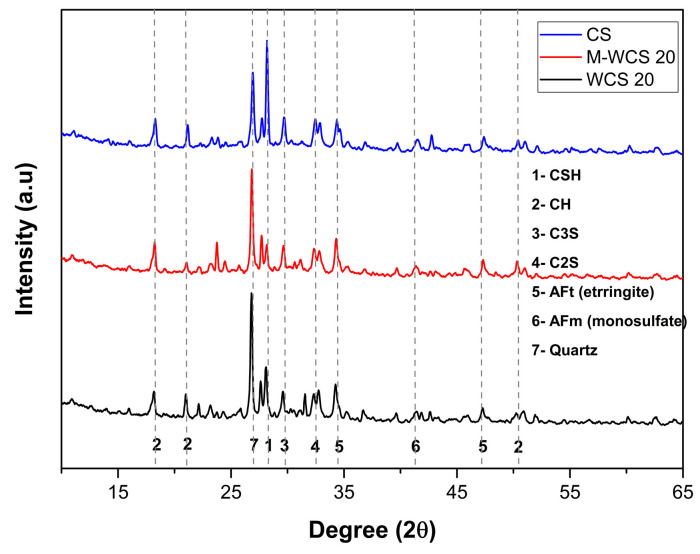
XRD spectrum of repair specimen.

**Figure 14 materials-17-02037-f014:**
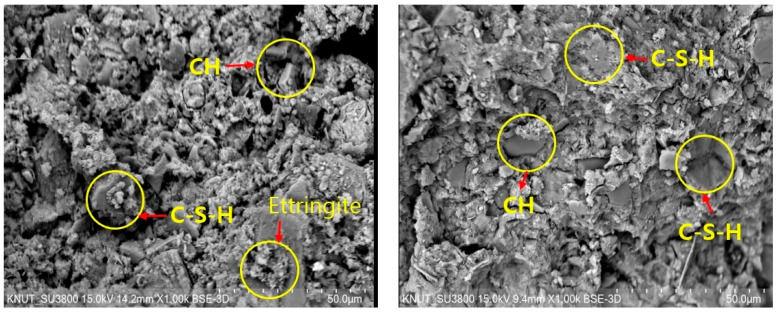
SEM analysis: (**a**) WCS 20 @ 3 days, (**b**) WCS 20 @ 28 days, (**c**) M-WCS @ 3 days, and (**d**) M-WCS @ 28 days.

**Table 1 materials-17-02037-t001:** Chemical composition and physical properties of OPC.

Oxide Content/wt.%						Physical Characteristics		
SiO_2_	Al_2_O	CaO	Fe_2_O_3_	MgO	K_2_O	Specific Gravity (g/cm^3^)	Surface Area (cm^2^/g)	Ig. Loss
20.8	6.3	62.0	3.2	2.9	2.1	2.19	5784	1.5

**Table 2 materials-17-02037-t002:** Chemical composition and physical properties of WWS.

Oxide Content/wt.%						Physical Characteristics	
SiO_2_	Al_2_O	CaO	Fe_2_O_3_	MgO	K_2_O	Specific Gravity (g/cm^3^)	Surface Area (cm^2^/g)
48.7	23.6	2.65	5.98	1.55	4.5	2.19	5784

**Table 3 materials-17-02037-t003:** Mixture proportion of repairing cement mortar.

Mix ID	W/B (%)		All Units Are (g)			
		Cement	WWS		Sand	WR (%)
			Treated	Non-Treated		
CS	40	700	-	-		
WCS 10		630	70	-		
WCS 20		560	140	-	1400	0.7
M-WCS 10		630	-	70		
M-WCS 20		560	-	140		

(WR) Water-reducer: wt.% of cement weight.

**Table 4 materials-17-02037-t004:** Dimension of specimens.

Test ID	Specimen Dimension (mm^3^)	Shape
Compressive test	50 × 50 × 50	prism
Flexural test	40 × 40 × 160	prism
Freeze–thaw test	100 × 100 × 400	prism
Chlorine penetration test	Φ 100 × 50	Cylinder (disc)
Carbonation depth test	Φ 100 × 50	Cylinder (disc)
Water absorption rate test	50 × 50 × 50	Cubic

**Table 5 materials-17-02037-t005:** Environmental hazard test measurement results.

Elements	Standard	Test Result
Pb	<3 mg/L	N/D (Non-Detected)
Cu	<3 mg/L	N/D (Non-Detected)
As	<1.5 mg/L	N/D (Non-Detected)
Cd	<0.005 mg/L	N/D (Non-Detected)
Cr^6+^	<1.5 mg/L	N/D (Non-Detected)
Hg	<0.3 mg/L	N/D (Non-Detected)
organic phosphorus	<1 mg/L	N/D (Non-Detected)

## Data Availability

Data are contained within the article.

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
