# Peer review of "Utilizing Wheel Washing Machine Sludge as a Cement Substitute in Repair Mortar: An Experimental Investigation into Material Characteristics"

_materials, 2024, doi:10.3390/ma17092037_

Round 1
Reviewer 1 Report
Comments and Suggestions for Authors
From my point of view, this text shows an important work done by the authors that could become an interesting article. However, in my opinion, this text is not acceptable to be published yet due to the following.
1-Introduction lacks some final lines briefly describing the structure of the paper for potential readers.
2- In section 2.2 it is unclear if the number of distinct specimens is 7 as presented in the text or 5 as depicted in tables 3 and 4. Also the total number of specimens is unclear. Although the referent standards are clear, the experimental champaign would be more understandable if the authors clarify the number of specimens types and the number of specimens tested of each type.
3- The types of tests performed is also unclear. Table 4 presents them but the text of section 2.2 lacks introducing the 5 types of tests. The types of tests in table 4 do not coincide properly with the types of tests presented in the subsections of 2.3 and the results. It is crucial that the number and names of tests coincide all through the article in order to ensure the paper rigor. The article also lacks a general explanation and justification of the types of tests performed. It could be either in Section 2.2 or 2.3, though the chosen section to do so will need to change the title accordingly.
4-The manuscript lacks discussing the results before drawing conclusions. This is comparing the results to similar former studies. To solve this issue, the reviewer suggests the authors to convert section 3 to a results and discussion section and add explicit comparisons of this research project results to former studies. This is crucial because will enable authors to backup conclusions such as the last concluding point with proper grounds.
5-The conclusions need rewriting because of several problems. The first sentence presents an objective beyond this research project objectives. The last paragraph present conclusions completely out of context that must be substituted by conclusions from this research project and within its boundaries. Conclusions should also describe limitations, for example regarding the environmental hazard analysis or most properties, could they change depending on the WWS origin?
Other comments
6- The first sentences of the introduction are strong statement that lack relying on references or prove.
7- General revision of spelling and writing is required. The captions of sub sections 2.3.2 and 2.3.3 need revision in the use of capitals, property or properties? Line 142 also the use of capital letters. Section 3.2 caption, line 257 caption, Section 3.3 caption, etc.
Comments on the Quality of English LanguageGeneral revision of spelling and writing is required. The captions of sub sections 2.3.2 and 2.3.3 need revision in the use of capitals, property or properties? Line 142 also the use of capital letters. Section 3.2 caption, line 257 caption, Section 3.3 caption, etc.
Author Response
Respected Reviewer,
Thank you for reviewing our manuscript entitled “Utilizing Wheel Washing Machine Sludge as a Cement Substitute in Repair Mortar: An Experimental Investigation into Material Characteristics” for possible publication in the journal “Materials”. We are thankful to you for your quick and valuable feedbacks to improve the quality of our manuscript for possible publication in the journal. We have revised the manuscript according to your comments and suggestions. Please see the response file.

Reviewer 2 Report
Comments and Suggestions for Authors
Dear Authors,
thank you for your manuscript, here will be following comments:
1. abstract is too general; please rewrite it and more specifically indicate your novelty and research outcome/results and conclusions;
2. Introduction doesn’t contain any data! Please add some comparison to earlier published results, some statistics, performance characteristics data, etc.
3. line 137: “flexural specimens”?! please cross check terminology
4. water absorption according which standard? please indicate for each test standards.
5. Figure4, please show your setup. Setup figure from standard is irrelevant.
6. Conclusion please add some data
Comments on the Quality of English Language
ok
Author Response

(The authors gave the same response as above.)

Round 2
Reviewer 1 Report
Comments and Suggestions for Authors
The article has improved compared to the previous version. Nevertheless, the end of the introduction lack some lines breifly describing the structure of the article and its main contents. The authors have added a second abstract of the article that is confusing, incomplete and unnecessary. Moreover, the presentation of the tests and specimens in 2.1-2.3 is still confusing. These sections still lack a general presentation in the body of the text of all the tests types that strictly coincides with 2.3 sub sections and tables.
Comments on the Quality of English LanguageThere are many problems in the use of capital letters, spelling... Lines 86, 93 (susch), 94, 289, 452, etc.
Author Response
Respected Reviewer,
Thank you for reviewing our manuscript entitled “Utilizing Wheel Washing Machine Sludge as a Cement Substitute in Repair Mortar: An Experimental Investigation into Material Characteristics” for possible publication in the journal of “Materials”. We are thankful to you for your quick and valuable feedback to improve the quality of our manuscript for possible publication in the journal. We have revised the manuscript according to your comments and suggestions. The pointwise replies are given here in the response file.

Reviewer 2 Report
Comments and Suggestions for Authors
no further remarks
Comments on the Quality of English Languageok
Author Response

(The authors gave the same response as above.)
